# Regulatory Role of Sphingosine-1-Phosphate and C16:0 Ceramide, in Immunogenic Cell Death of Colon Cancer Cells Induced by Bak/Bax-Activation

**DOI:** 10.3390/cancers14215182

**Published:** 2022-10-22

**Authors:** Jeremy A. Hengst, Asvelt J. Nduwumwami, Jong K. Yun

**Affiliations:** 1Department of Pharmacology, Pennsylvania State University College of Medicine, 500 University Drive, Hershey, PA 17033, USA; 2Department of Experimental Radiation Oncology, The University of Texas MD Anderson Cancer Center, 6565 MD Anderson Blvd, Houston, TX 77030, USA

**Keywords:** sphingolipids, ceramide, sphingosine phosphate, cell signaling, inflammation, immunogenic cell death, Bak, Bax, colorectal cancer

## Abstract

**Simple Summary:**

Recent advances in our understanding of the immune response to tumors highlighted the importance of immunogenic cell death (ICD) as a novel therapeutic strategy. This study aims to elucidate the role(s) of sphingolipids in the colorectal cancer cell (CRC) response to ICD-inducing agents and demonstrates that sphingolipids are key regulators of ICD. Our findings provide insights into the cellular signaling mechanism of ICD that will lead to the development of improved therapeutic strategies that harness the power of the sphingolipids in the fight against CRC.

**Abstract:**

We recently identified the sphingosine kinases (SphK1/2) as key intracellular regulators of immunogenic cell death (ICD) in colorectal cancer (CRC) cells. To better understand the mechanism by which SphK inhibition enhances ICD, we focused on the intracellular signaling pathways leading to cell surface exposure of calreticulin (ectoCRT). Herein, we demonstrate that ABT-263 and AZD-5991, inhibitors of Bcl-2/Bcl-X_L_ and Mcl-1, respectively, induce the production of ectoCRT, indicative of ICD. Inhibition of SphK1 significantly enhanced ABT/AZD-induced ectoCRT production, in a caspase 8-dependent manner. Mechanistically, we demonstrate that ABT/AZD-induced Bak/Bax activation stimulates pro-survival SphK1/sphingosine-1-phosphate (S1P) signaling, which attenuates ectoCRT production. Additionally, we identified a regulatory role for ceramide synthase 6 (CerS6)/C16:0 ceramide in transporting of ectoCRT to the cell surface. Together, these results indicate that the sphingolipid metabolic regulators of the sphingolipid rheostat, S1P and C16:0 ceramide, influence survival/death decisions of CRC cells in response to ICD-inducing chemotherapeutic agents. Importantly, SphK1, which produces S1P, is a stress-responsive pro-survival lipid kinase that suppresses ICD. While ceramide, produced by the inhibition of SphK1 is required for production of the cell surface marker of ICD, ectoCRT. Thus, inhibition of SphK1 represents a means to enhance the therapeutic efficacy of ICD-inducing agents.

## 1. Introduction

Sphingosine-1-phosphate (S1P) is a pro-mitogenic/pro-survival sphingolipid with pleotropic effects on cancer cell growth, survival, metastasis and chemotherapeutic resistance. Targeting the enzymes responsible for S1P production, sphingosine kinase 1 and 2 (SphK1 and SphK2) has been extensively investigated pre-clinically as well as in human clinical trials. As a stand-alone therapy, SphK inhibition may have some therapeutic benefit, but as an adjuvant to standard chemo- or immuno-therapies, SphK inhibition has great potential. Indeed, a recent study has demonstrated that SphK1 inhibition enhances the effectiveness of immune checkpoint inhibitors through modulation of PGE2 synthesis in the tumor microenvironment (TME) [1].

While modulation of immunosuppression in the TME suggests a potential therapeutic benefit for SphK inhibition as part of the armament against cancer, we recently discovered a previously unrecognized role for sphingolipids as regulators of cancer cell immunogenicity [2]. To this end, we demonstrated that inhibition of SphK1 and/or SphK2 and therefore, S1P production, leading to ceramide (Cer) accumulation enhances the ability of certain chemotherapies (e.g., mitoxantrone) to induce a specific form of regulated cell death termed immunogenic cell death (ICD).

Historically, the dogmatic view of the mechanism-of-action of cancer chemotherapies has been that they induce apoptosis, a tolerogenic form of regulated cell death, and hence remain hidden from the innate immune system. However, recent studies have challenged this assertion by demonstrating that ICD-inducing agents including mitoxantrone prompt cancer cells to release adjuvants, termed danger-associated molecular patterns (DAMPs) in a coordinated fashion to induce activation of the innate/adaptive immune system [3]. In contrast, chemotherapeutic agents that are unable induce ICD fail to promote DAMP release. Nevertheless, the understanding of the intracellular pathways leading to ICD is not nearly as well understood as the pathways leading to apoptotic cell death.

The current understanding of ICD is based primarily on the mechanism by which mitoxantrone (MTX) induces ICD [4]. MTX-induced endoplasmic reticulum (ER) stress leads to activation of the protein kinase R-like endoplasmic reticulum kinase (PERK) which, in turn, mediates phosphorylation of eukaryotic initiation factor 2α (eIF2α), a process thought to be pathognomonic to ICD [5]. Through multiple intracellular processes, eIF2α phosphorylation induces production of several DAMPs including exposure of the normally ER resident chaperone, calreticulin (CRT), at the PM surface (ectoCRT), the autophagy-mediated secretion of ATP to the extracellular milieu, and the extracellular release of high-mobility group box 1 (HMGB1). ER stress is transmitted to activation of caspase 8, cleavage of BAP31 and activation of Bak/Bax at the ER membrane. Upon Bak/Bax activation, CRT is translocated to the PM surface in a process requiring SNAREs. For a review of the current understanding of ICD, see Galluzzi et al., 2020 [3].

Our recent discovery of the role of sphingolipids, including Cer, in the process of ICD prompted us to reexamine the intracellular mechanism of ICD. In particular, we focused on the role(s) of Bak/Bax in ICD. Bak/Bax activation and Cer induced cell death have been associated in numerous studies [6,7]. In addition to their better-accepted role in intrinsic/extrinsic apoptosis associated with mitochondrial localization, Bak/Bax are also localized to the ER [8]. Indeed, Bak/Bax activation at the ER has been associated with depletion of Ca^2+^ from ER stores [9,10,11,12,13]. Similarly, depletion of ER Ca^2+^ has been linked to ectoCRT exposure [14]. Thus, we examined whether direct activation of Bak/Bax could induce ICD and what role sphingolipids played in this process.

Herein, we report that at very low concentrations, inhibitors of pro-survival Bcl-2 family proteins (ABT-263 and AZD-5991 in combination), by themselves, do not induce ICD. However, these concentrations are sufficient to prime cells to undergo ICD when sphingolipid metabolism is blocked by inhibiting SphK1, which also does not induce ICD by itself. Thus, SphK1-generated S1P is a lipid metabolite that is critical for cancer cell survival under conditions that promote endoplasmic reticulum stress. We further elucidated the mechanism by which SphK1/S1P regulates the cellular response to ER stress and have determined that stabilization of cellular FLICE (FADD-like IL-1β-converting enzyme)-inhibitory protein (c-FLIP) is a key pro-survival function of S1P. Moreover, we have also identified C16:0 Cer as a specific Cer species required for transport of DAMPs from their intracellular locales to the cell surface/extracellular environment, which is the hallmark of ICD.

## 2. Materials and Methods

### 2.1. Compounds and Antibodies

ABT-263, AZD5991, Mitoxantrone, PF-543, Thapsigargin, KIRA6, GSK2606414, Necrostatin-1, Disulfiram and z-IETD-fmk were all purchased from Selleckchem (Houston, TX, USA). Fumonisin B1 was from Cayman Chemicals (Ann Arbor, MI, USA). All compounds were prepared in a DMSO vehicle. Anti-Calreticulin, anti-phospho eIF2α Ser51, anti-full length Caspase 8, anti-Bap31, anti-Cleaved PARP, anti-GRP78/BiP, anti-Bcl-XL, anti-FLIP, anti-full length BID, and anti-total SphK1 antibodies were from Cell Signaling Technologies (Beverly, MA, USA). Anti-GAPDH, and anti-CerS6 antibodies were from Santa Cruz Biotechnologies (Dallas, TX, USA). Anti-phospho-SphK1 Ser225 antibodies were from ECM Biosciences (Versailles, KY, USA). Anti-CerS5 antibodies were from LSBio (Seattle, WA, USA).

### 2.2. Cell Lines and Culture Conditions

Human colorectal DLD-1 (CCL-221) and human acute myeloid leukemia THP-1 (TIB-202) cells were obtained from ATCC, (Manassas, VA). HEK293 cells over-expressing FLAG-SphK1 and GFP-SphK1 were described previously [15,16]. All cells were cultured at 37 °C in a humidified atmosphere of 5% CO_2_ in Dulbecco’s Modified Eagle Medium (DMEM) supplemented with 10% fetal bovine serum (FBS) and penicillin/streptomycin.

### 2.3. Crispr/Cas9 Knock-Out of CerS5 and CerS6 in DLD-1 Cells

CerS5 and CerS6 knock-out clones of DLD-1 cells were generated using the LentiCRISPR v2.0 system with guide RNAs (gRNAs; CerS5; 5′-GCTTGTCCTGATTCCTCCGA-3′ and CerS6; 5′-GGCTCCCGCACAATGTCACC-3′) purchased from GenScript (Piscataway, NJ, USA). DLD-1 cells were transiently transfected using Lipofectamine 2000. After 48 h, cells were selected in puromycin (10 ug/mL) for 24 h and maintained in 1 ug/mL puromycin until single clones were isolated. Screening for CerS protein expression knock-out was conducted by Western blot analysis using the appropriate CerS antibodies.

### 2.4. Detection of Cell Surface CRT

Analysis of cell surface exposure of CRT was described previously [2]. Briefly, cells were seeded at about 3 × 10^5^ cells/well in six well plates in complete DMEM for 24 h, then transferred to DMEM containing 5% FBS and penicillin/streptomycin in the presence of treatments for 48 h. Cells were collected by trypsinization, followed by three washes in PBS containing 2% FBS, stained with anti-calreticulin (D3E6) phycoerythrin (PE) conjugated antibody at 4 °C for 1 h, washed as above, and CRT was detected by Muse cell analyzer (SmartFlare detection settings).

### 2.5. Sphingolipid Analysis

DLD-1 cells were treated with ABT/AZD (0.5 µM and 0.25 µM, respectively, for low dose and 1.0 µM and 0.5 µM, respectively, for high dose) and/or PF-543 (5 μM) for 24 h. Individual cell samples were collected by trypsinization, pelleted and washed with PBS and flash frozen. Sphingolipidomic analysis was conducted by the Lipidomic Shared Resource Facility (Medical University of South Carolina, Charleston, SC, USA). Appropriate lipid standards were employed per standard Lipidomic Shared Resource Facility protocols. Sphingolipid levels are expressed as pmoles of sphingolipid per µg of total protein.

### 2.6. Whole Cell Lysate Preparation

Total cell lysate was obtained by incubating treated and untreated DLD-1 cells in 1X RIPA (Cell Signaling Tech, Beverly, MA, USA), with phosphatase inhibitor cocktail and protease inhibitor tablets (Roche), for 30 min at 4 °C and followed by removal of cell debris by centrifugation at 20,000× *g* at 4 °C. Protein concentrations were determined by BCA Assay (Pierce, Waltham, MA, USA).

### 2.7. Phagocytosis Assays

THP-1 acute myeloid leukemia cells were differentiated into M1 macrophages according to the protocol of Park et al., [17]. DLD-1 cells were treated for 24 h and labeled with CellTrace CSFE (Invitrogen). THP-1 macrophages were labeled with CellTracker Far Red (Invitrogen). 1.0 × 10^5^ DLD-1 and THP-1 cells were co-cultured and visualized by time-lapsed fluorescent microscopy for 18 h (Keyence BX-710).

### 2.8. Vaccination Assays

Animal studies were approved by the Penn State Hershey IACUC Committee. Mice (*n* = 6/group) were initially vaccinated with 1.0 × 10^6^ injured/dead MC-38 CRC cells treated with mitoxantrone (MTX; 0.3 µM) alone or the combination of ABT/AZD + PF-543 (AAPF; 0.06 µM/0.13 µM/2.5 µM) for 24 h. Vehicle control cells were subjected to repeated freeze/thaw cycles. 1 week later, vaccinated mice were challenged with live 4.0 × 10^5^ MC-38 cells by injection into the contralateral flank. Kaplan–Meier survival curves were generated for each treatment group.

### 2.9. Statistical Analysis

Where appropriate, statistical analysis was performed using one-way ANOVA followed by Tukey’s multiple comparison test or two-way ANOVA using GraphPad PRISM. Results are reported as average values among replicates with 95% CI ranges determined by Tukey’s test. Results are considered significant when the *p* value is less than 0.05. For vaccination assays, Log-Rank (Mantel-Cox) tests were performed using GraphPad PRISM.

## 3. Results

### 3.1. ABT-263 Induces and AZD-5991 Enhances Cell Surface Exposure of Calreticulin (ectoCRT)

ER stress and ER Ca^2+^ depletion have been implicated in the production of cell surface CRT (ectoCRT) and activation of the pro-apoptotic Bcl-2 family members, Bak and Bax, have been identified as downstream factors required for ectoCRT exposure [4]. Bak and Bax have been localized to the ER membranes in addition to the mitochondrial membrane [8]. When activated by BH3-only proteins, Bak and Bax oligomerize to form channels that induce mitochondrial outer membrane permeability (MOMP) [18]. We reasoned that activation of ER resident Bak and Bax may, similarly, induce ER membrane permeabilization and ER Ca^2+^ depletion, without causing ER stress, and that their activation may be sufficient to induce ectoCRT exposure. We, therefore, assessed the ability of agents that activate these pro-apoptotic proteins to cause CRT cell surface exposure in DLD-1 cells. Bak and Bax are held inactive by Bcl-XL and Bcl-2, two of the anti-apoptotic Bcl-2 family proteins. Navitoclax (ABT-263; ABT) a BH3-mimetic inhibits the interactions between Bcl-2/XL with Bak and Bax thereby activating the latter [19]. Oligomerization and activation of Bak can also be prevented by its interaction with Mcl-1, the third anti-apoptotic Bcl-2 family member. AZD-5991 (AZD), another BH3-mimetic, has been shown to selectively bind Mcl-1 thereby resulting in Bak activation [20].

We treated DLD-1 cells with either ABT or AZD and determined ectoCRT exposure by flow cytometry (Figure 1A) as previously demonstrated [2]. ABT dose dependently increased ectoCRT exposure. In contrast to ABT, AZD did not induce ectoCRT even at 3 μM, a concentration greater than 100× its IC_50_ value [20]. 3 µM ABT significantly increased ectoCRT exposure relative to 1 µM ABT from an average mean fluorescent intensity (MFI) of 42.5 to 113.7 (95% CI 40.3 to 102.0, *p* = 0.0004) Similarly, as shown in Figure 1B, combining ineffective doses of ABT (0.5 µM) and AZD (0.5 µM) significantly enhanced the average fold change in mean fluorescent intensity (referred to as average MFI throughout) of ectoCRT exposure relative to either treatment alone. The MFI increased from 2.067 for 0.5 µM AZD alone and 24.83 for ABT alone (AZD versus ABT, *p* = 0.0010) to 197.1 for 0.5 µM ABT + 0.5 µM AZD, (95% CI of AZD alone (2.067) versus AZD + ABT 125.3 to 268.9; *p* < 0.0001), (95% CI of ABT alone versus AZD + ABT 100.5 to 244.0; *p* = 0.0003). These data show that ABT induces ectoCRT and that disrupting the Mcl-1:Bak interaction enhances ectoCRT displayed at the surface of DLD-1 CRC cells.

### 3.2. Sphingosine Kinase Inhibition Enhances the Production of ectoCRT by Minimally-Effective Doses of ABT/AZD

We have previously demonstrated that inhibition of SphK1/2 enhances ectoCRT production by the known ICD-inducing agent mitoxantrone MTX through accumulation of Cer [2]. Bak, as well as Bax have been associated with induction of Cer synthesis through multiple studies [6,7,21]. Cer has also been shown to form channels that are functionally regulated by the pro- and anti-apoptotic Bcl-2 family proteins [22,23]. Moreover, as mentioned above Bak and Bax have been shown to form pores, albeit in the mitochondrial outer membrane [18]. Given that the ER is the site of Cer synthesis, we reasoned that activation of Bak and Bax might be enhanced by inhibition of the SphKs. In this scenario, SphK inhibition would block the conversion of Sph to S1P leading to the accumulation of Cer species either through enhanced utilization of Sph by the Cer synthases to form Cer species or through inhibition of Cer degradation. Cer accumulation, in turn leads to the formation of channels/pores in the ER stabilized by Bak/Bax inducing ER Ca^2+^ leakage and induction of ICD.

As shown in Figure 2A, we identified concentrations of ABT and AZD that, by themselves, had no effect on production of ectoCRT, nor did they induce ectoCRT in combination. As we observed previously, the SphK inhibitor PF-543 (Pfizer) alone has no effect on production of ectoCRT. At these concentrations, however, the addition of PF-543 significantly and dose-dependently induced ectoCRT exposure as measured by flow-cytometry [2]. The average MFI increased from 1.9 for ABT + AZD (AA) to 9.7–32.75 (1.25 µM PF-543—20 µM PF-543) for AA + PF-543 (95% CI of AA versus AA + PF-543 (1.25 µM) 2.13 to 13.5 *p* = 0.0055) all other PF-543 concentration *p* < 0.0001).

The mechanism of action of MTX-induced ectoCRT exposure has been shown to require the phosphorylation/inactivation of eukaryotic Initiating Factor 2 alpha (eIF2α), activity of Caspase 8 (Casp8) and cleavage of the Casp8 substrate Bap31 (28 kDa) yielding a 20 kDa fragment [4,5]. To determine whether ABT/AZD induced ectoCRT exposure occurred by the same mechanism as MTX-induce ectoCRT exposure, we next compared the effects of ABT/AZD ± PF-543 to that of MTX ± PF-543. As indicated in Figure 2B, ABT/AZD (0.5 µM/0.25 µM respectively; higher than Figure 2A) induced minimal phosphorylation of eIF2α as compared to MTX (1 µM or 4 µM) alone. PF-543 did increase phosphorylation of eIF2α in response to ABT/AZD treatment, however, the effect was not as great as that observed in combination with 1 µM MTX.

We next examined activation of Casp8 and cleavage of the substrate Bap31. PF-543 alone did not induce activation of Casp8. ABT/AZD and MTX (1 µM) alone induced minimal activation of Casp8 as judged by the partial cleavage of Bap31. 4 µM MTX alone enhanced the activation of Casp8 and increased Bap31 cleavage. The addition of PF-543 (5 µM) significantly enhanced the activation of Casp8 by both ABT/AZD and MTX (1 µM), resulted in at least 50% conversion of Bap31 to the cleaved 20 kDa product and increased generation of the cleaved form of PARP. GRP78/BiP and Bcl-XL were employed as a loading controls and were not affected by Casp8 activation. Interestingly, ABT/AZD + PF-543 induced full activation of Casp8, yet produced the lowest levels of phosphorylated eIF2α suggesting that eIF2α phosphorylation is not directly correlated to the extent of ICD induction, at least for ABT/AZD. This is consistent with the idea that acting directly on Bax and Bak can circumvent the requirement for induction of ER stress attributed to other ICD-inducing agents, such as MTX.

We recently demonstrated that ectoCRT is present at the lipid raft fraction of the plasma membrane as a disulfide-linked dimer, in response to MTX ± PF-543 [2]. To ensure that ABT/AZD also induced disulfide-linked dimerization of ectoCRT, we assayed for the presence of dimeric CRT in whole cell lysates of DLD-1 cells, treated as above, under non-reducing conditions. As shown in Figure 2C, consistent with the activation of Casp8/Bap31 cleavage ABT/AZD induced and PF-543 enhanced CRT dimerization.

ABT/AZD alone are minimally effective at induction of ectoCRT exposure. One possible explanation for this is that ABT/AZD do partially activate Bak/Bax to induce minimal pore formation and Ca^2+^ leakage from the ER, but the activity of ER Ca^2+^ pumps (i.e., SERCA) are sufficient to maintain a ER: cytosol Ca^2+^ gradient preventing dimerization of cytosolic CRT and transport to the cell surface. To test this possibility, we next employed the SERCA inhibitor thapsigargin to prevent Ca^2+^ reentry into the ER in the presence of ABT/AZD at low doses. As shown in Figure 2D, ABT, AZD and their combination, again, had minimal effect on ectoCRT production. The SERCA inhibitor, thapsigargin, by itself had minimal effect. However, in combination with low dose ABT/AZD inhibition of the cells ability to maintain ER Ca^2+^ homeostasis significantly enhanced the effects of ABT/AZD. Average MFI increased from 1.9 for ABT/AZD to 10.2 (0.5 µM thapsigargin) and 27.5 (1.0 µM thapsigargin) (95% CI 3.1 to 13.5, *p* = 0.0054 for 0.5 µM thapsigargin and 20.4 to 30.8, *p* < 0.0001 for 1.0 µM thapsigargin). This implies that ABT/AZD acts at the ER membrane to induce Ca^2+^ leakage from the ER to the cytosol triggering ectoCRT production. Consistent with this, thapsigargin enhanced the ABT/AZD-induced dimerization of CRT (Figure 2E).

### 3.3. SphK Inhibitors Synergize with ABT/AZD to Induce c-FLIP_L/S_ Depletion Upstream of Caspase 8 Activation

Casp8 activation is regulated by proteolytic processing through autocatalytic- or cross-processing and by the presence of a ubiquitously expressed endogenous inhibitor, cellular FLICE inhibitory protein (c-FLIP) [24]. How Casp8 is activated intracellularly during ICD is not clear. Thus, we examined whether levels of c-FLIP were affected by ABT/AZD ± PF-543. As shown in Figure 3A, low dose ABT/AZD had minimal effect on levels c-FLIP and PF-543 alone did not affect c-FLIP stability. In combination, however, ABT/AZD + PF-543 almost completely abolished c-FLIP levels in DLD-1 cells leading to Casp8 activation cleavage of the Casp8 substrates Bap31 and BID. Consistent with its role as an inhibitor of Casp8 and therefore upstream of Casp8, inhibition of Casp8 did not affect c-FLIP levels although it did completely block cleavage Bap31 and BID.

Inhibition of Casp8 attenuated formation of CRT dimers induced by ABT/AZD alone and in combination with PF-543 (Figure 3B). Interestingly, previous studies have suggested that Caspase 8 activation occurs upstream of Bak/Bax activation and that Bak/Bax exert their effects at the mitochondria [4]. Figure 3A,B indicate that Bak/Bax activation may be upstream of c-FLIP depletion and Caspase 8 activation. If this were the case, then Caspase 8 inhibition should attenuate ectoCRT exposure induced by ABT/AZD alone and in combination with PF-543. Indeed, as shown in Figure 3C, inhibition of Caspase 8 dose dependently reduced ectoCRT exposure induced by ABT/AZD alone and in combination with PF-543. Average MFI decreased from 415.9 to 155.7 (10 µM Casp8inh) and 66.1 (20 µM Casp8inh) (95% CI for ABT/AZD + PF-543 versus ABT = AZD + PF-543 + 10 µM Casp8inh, 171.5 to 348.8 *p* < 0.0001). These results are consistent with the hypothesis that Bak/Bax induced activation of Caspase 8 occurs at the ER and is enhanced by alteration of sphingolipid levels through inhibition of SphK1 activity.

### 3.4. SphK1/S1P Stabilizes c-FLIP and Prevents ABT/AZD Induced Caspase 8 Activation

Early studies in yeast proposed that SphK was a pro-survival, stress-responsive protein [25]. More recent studies demonstrated that SphK1 activity is enhanced by ER stress in keratinocytes [26,27]. We hypothesized that ER-stress responsive SphK1 activation protects CRC cells from chemotherapeutic agent-induced ICD. To test this hypothesis, we established a DLD-1 cell line over-expressing a GFP-SphK1 fusion construct, described previously [16], (GFP-SphK1; Figure 4A). As shown in Figure 4B, both PF-543 and ABC294640 significantly enhanced ABT/AZD (AA)-induced ectoCRT expression in wild-type DLD-1 cells. In contrast, in SphK1 over-expressing DLD-1 cells, ectoCRT expression was significantly reduced relative to wild-type DLD-1 cells in response to ABT/AZD ± PF-543 or ABC294640. The average MFI between wild-type DLD-1 and GFP-SphK1 expressing DLD-1 cells decreased from 353.8 to 100.8 for ABT/AZD alone (95%CI 207.6 to 298.5, *p* < 0.0001), from 644.7 to 199.5 (95% CI 399.7 to 490.6, *p* < 0.0001) for ABT/AZD + PF-543 and from 555.2 to 160.7 (95% CI 349.1 to 440.0, *p* < 0.0001) for ABT/AZD + ABC294640. This finding is consistent with our hypothesis that S1P produced by SphK1 activation inhibits ICD by stabilizing c-FLIP/inhibits caspase 8 activity and ectoCRT production.

Notably, PF-543 and ABC294640 are reportedly selective inhibitors of SphK1 and SphK2, respectively. However, our previous studies measuring direct target engagement of both inhibitors indicate that neither inhibitor is selective at micromolar concentrations [15]. To this point, we have used PF-543 and ABC294640 in the low micromolar concentration range (2.5–5 µM) in 5% FBS/DMEM, and therefore the relative contributions of SphK1 vs. SphK2 are not conclusive at this time. In vitro biochemical studies of PF-543, including ours [15], indicate that its IC_50_ for direct target engagement of SphK1, in whole cells, is in low nM range and is approximately 1.25–2.5 µM for SphK2 [28]. That micromolar concentrations of PF-543 were required to enhance induction of ectoCRT exposure suggested that both SphK1 and SphK2 must be inhibited.

However, because it is known that S1P is present in FBS at high concentrations, we reasoned that it is possible that S1P derived from the cell culture medium contributed to its’ pro-survival effects against ABT/AZD-induced cell surface exposure of ectoCRT. To this end, S1P binds to and activates a cohort of five cell surface receptors (S1PR1-S1PR5; [29]). Additionally, it has recently been reported that serum S1P can be directly imported into the cells by the Spns2 and MFSD2B import/export pumps [30], suggesting that in 5% FBS, S1P can enter the cell to counteract the intracellular reduction of S1P, by SphK inhibition in combination with ABT/AZD. To address this concern, we delipidated FBS according to previously established protocols [31]. Replacing 5% FBS with 5% delipidated FBS in the cell culture medium, our results show that 100 nM PF-543, as well as 100 nM ABC294640, were sufficient to enhance ABT/AZD induced cell surface exposure of ectoCRT relative to ABT/AZD alone (Figure 4C). SphK1 over-expression again significantly reduced ectoCRT expression relative to wild-type DLD-1 cells in response to ABT/AZD ± PF-543 or ABC294640. The average MFI between wild-type DLD-1 and GFP-SphK1 expressing DLD-1 cells decreased from 452.0 to 127.3 for ABT/AZD alone (95%CI 297.8 to 352.5, *p* < 0.0001), from 623.8 to 142.8 (95% CI 454.1 to 507.8, *p* < 0.0001) for ABT/AZD + PF-543 and from 462.6 to 128.2 (95% CI 307.6 to 361.3, *p* < 0.0001) for ABT/AZD + ABC294640. This indicates that S1P produced intracellularly by SphK1 mediates the resistance of DLD-1 cells to ABT/AZD-induced cell surface exposure of ectoCRT.

In support of this, GFP-SphK1 over-expressing DLD-1 cells are significantly more resistant to ABT/AZD alone and in combination with PF-543 in 5% delipidated FBS (Figure 4C). Mechanistically, ABT/AZD + PF-543 induced c-FLIP depletion in wild-type DLD-1 cells (Figure 4D), but not in GFP-SphK1 cells, indicating a role of SphK1 in resistance to caspase 8-dependent ICD. Furthermore, we did not detect the formation of dimeric CRT in response to ABT/AZD alone or in combination with PF-543 in the SphK1 over-expressing cells (Figure 4D) suggesting that c-FLIP depletion is required for formation of dimeric CRT.

### 3.5. Cer Is Required for Cell Surface Exposure of ectoCRT

To better understand whether inhibition of S1P formation or accumulation of Cer might be responsible for the effects of ABT/AZD + PF-543 on c-FLIP stability, we employed the Cer synthase inhibitor, fumonisin B1 (FB1) in wild-type DLD-1 cells. As shown in Figure 5A, ABT/AZD + PF-543 induced c-FLIP depletion and Bap31 cleavage. The addition of FB1 did not affect ABT/AZD + PF-543 induced c-FLIP depletion or Bap31 cleavage, indicating that synthesis of Cer is not required for depletion of c-FLIP induced by inhibition of SphK1. We further examined the effects of FB1 on dimerization of CRT and observed that FB1 enhanced the formation of dimeric CRT, in whole cell lysates, by ABT/AZD + PF-543 (Figure 5B).

The enhancement of CRT dimer formation implies that cell surface exposure of ectoCRT would also be increased by FB1 treatment. However, as shown in Figure 5C, FB1 significantly attenuated the cell surface exposure of ectoCRT. The average MFI decreased from 292.0 for ABT/AZD + PF-543 to 57.4 with the addition of FB1 (95% CI 185.5 to 283.6, *p* < 0.0001). One possible explanation for this observation would be that translocation of dimeric CRT from the intracellular space to the plasma membrane is altered by inhibition of Cer synthesis. The increase of dimeric CRT under conditions of ABT/AZD + PF-543 + FB1 treatment, relative to ABT/AZD + PF-543 alone, could indicate that in addition to cell surface exposure of ectoCRT, a portion of ectoCRT is shed into the culture medium, either directly or inside of exocytic vesicles. Inhibition of transport to the cell surface would block the shedding process resulting in more dimeric CRT retained intracellularly.

Together, these data indicate that inhibition of SphK1 enhances the depletion of c-FLIP leading to Casp8 activation. This suggests that SphK1 has a pro-survival/anti-ICD role to overcome ER stress and prevent ectoCRT production. Furthermore, because cell surface exposure of ectoCRT was attenuated by inhibition of Cer synthesis, this implies that Cer is an important factor required for cell surface exposure of ectoCRT. Thus, sphingolipids appear to have biphasic effects as both anti- (S1P) and pro- (Cer) ICD sphingolipids, consistent with their better understood roles in apoptotic cell death.

### 3.6. Inhibition of Inositol Requiring Enzyme 1 Alpha (IRE1α) Blocks Activating Phosphorylation of SphK1 and Enhances ectoCRT Production by ABT/AZD

To better elucidate the role of SphK1/S1P as an inhibitor of the events leading to ICD, we next examined the regulation of SphK1 activity. SphK1 has a constitutive basal activity that is enhanced by phosphorylation at Ser225. To date, Erk1/2 is the only kinase known to phosphorylate SphK1 in response to growth factor stimulation (e.g., TNFα and phorbol esters) [32]. Thus, we considered the role of Erk1/2 as an activator of SphK1. Preliminary studies indicated that inhibition of Erk1/2 had no effect on the phosphorylation of SphK1 and Ser225 in HEK293 cells over-expressing SphK1.

If Erk1/2 were not responsible for SphK1 activation, we next considered the possible role of kinases involved in ER stress and the unfolded protein response (UPR). Interestingly, SphK1 has recently been shown to form a complex with TRAF2 and IRE1α in response to ER stress [26]. IRE1α is one of the three major arms of the UPR pathway and it possesses both protein kinase and RNase activities. To date, it is thought that IRE1α autophosphorylation is the only activity of the IRE1α kinase domain. We tested whether inhibition of IRE1α kinase activity would affect basal phosphorylation of SphK1 in the HEK293 over-expression system. To our surprise, inhibition of IRE1α kinase activity with KIRA6 completely abrogated basal SphK1 phosphorylation as low as 2.5 µM while inhibition of Erk1/2, using PD98059 (10 µM) had no effect of phosphorylation of SphK1 at Ser225 using anti-phospho Ser225 antibodies (Figure 6A). Inhibition of IRE1α did not affect levels of total SphK1, implying that IRE1α or a down-stream interacting kinase was responsible for the bulk phosphorylation of SphK1.

Bak and Bax have been shown to interact with and activate IRE1α [8]. This is consistent with the hypothesis that ABT/AZD could activate SphK1, via IRE1α, to counter-balance the pro-ICD inducing effects of ABT/AZD and thus, inhibition of SphK1 would synergistically/additively enhance the production of ectoCRT by ABT/AZD. We therefore, examined the effects of IRE1α inhibition on ectoCRT production by ABT/AZD ± PF-543. As shown in Figure 6B, inhibition of IRE1α did not induce ectoCRT production alone or in combination with PF-543 demonstrating that insult to the ER (i.e., ER stress or ER Ca^2+^ release) is required for ectoCRT production. ABT/AZD alone, at the concentrations employed, had no effect on ectoCRT production. However, inhibition of IRE1α (KIRA6) significantly enhanced ectoCRT production. The average MFI increased from 88.3 for ABT/AZD alone to 549.5 with the addition of KIRA6 (95% CI 375.8 to 546.5, *p* < 0.0001). The addition of KIRA6 to ABT/AZD+ PF-543 did not further enhance ectoCRT production consistent with the hypothesis that IRE1α is up-stream of SphK1 activity and is possibly directly responsible for SphK1 activation.

Interestingly, inhibition of another UPR kinase (protein kinase regulated by RNA (PKR)-like ER kinase (PERK)), one of four kinases that phosphorylate eIF2α at Ser51, also enhanced ectoCRT production by ABT/AZD. The average MFI increased from 88.3 for ABT/AZD alone to 535.3 with the addition of GSK2606414 (95% CI 361.6 to 532.3, *p* < 0.0001). This is in direct opposition to the effect of PERK inhibition on MTX-induced ectoCRT production. As with IRE1α, PERK inhibition did not further enhance the effects of ABT/AZD + PF-543 on ectoCRT production. These data are consistent with the role of ER stress in induction of ectoCRT/ICD, however, other forms of regulated cell death including necroptosis and pyroptosis have been reported to be immunogenic. To preclude the possibility that either of these forms of cell death were induced by ABT/AZD, we employed the necroptosis inhibitor, necrostatin-1 and the pyroptosis inhibitor, disulfiram and examined their effects on ABT/AZD + PF-543 induced ectoCRT production. Inhibition of these regulated cell death pathways had no effect on production of ectoCRT by ABT/AZD + PF-543 consistent with the induction of *bona fide* ICD by targeting Bak/Bax and SphK1.

### 3.7. ABT/AZD Induces Cer Accumulation Only at High Concentrations

We previously observed an increase in multiple Cer species in response to combined treatment of MTX + PF-543 in DLD-1 cells [2]. We believe that the observed increase occurs due to ER stress stimulation of CerS 1–6 activity in combination with the accumulation of Sph arising from inhibition of SphK1. Our results above suggest that low concentrations of ABT/AZD may not directly induce ER stress, per se, but may instead induce ER Ca^2+^ release. Other studies have demonstrated that IC_50_ concentrations of ABT-263 can directly stimulate the production of C16:0 Cer through Bak-mediated activation of CerS activity [7]. Thus, we performed sphingolipidomic analysis of DLD-1 cells with ABT/AZD (subIC_50_) alone and in combination with PF-543. As shown in Figure 7A, PF-543 alone induced the expected decrease in S1P/dhS1P levels and concomitantly induced an increase in Sph/dhSph levels. dhSph levels increased 2.2-fold with PF-543 treatment relative to vehicle (95% CI 1.22 to 3.24, *p* < 0.0001). Sph levels did not significantly change with PF-543 alone. The addition of ABT/AZD further enhanced dhSph levels (4.1-fold relative to vehicle; 95% CI 3.13 to 5.15, *p* < 0.0001) and significantly increased Sph levels relative to vehicle (1.4-fold, 95% CI 0.39 to 2.41, *p* = 0.0026). Low concentrations of ABT/AZD alone did not alter dhSph/Sph levels. Higher doses of ABT/AZD significantly enhanced both dhSph (3.0-fold, 95% CI 1.99 to 4.00, *p* < 0.0001) and Sph levels (2.3-fold, 95% CI 1.29 to 3.31, *p* < 0.0001) relative to vehicle.

PF-543 alone and low doses of or ABT/AZD alone had minor effects on Cer levels consistent with the fact that PF-543 is non-cytotoxic and that low doses of ABT/AZD were ineffective, as demonstrated above [28]. In combination with PF-543, ABT/AZD (low) induced increases in C18:0 (1.5-fold, 95% CI 1.69 to 2.92, *p* < 0.0001) and C24:1 Cer (1.6-fold, 95% CI 1.03 to 2.26, *p* < 0.0001), however, C14:0 and C16:0 Cer levels were not altered relative to vehicle treatment (Figure 7B). As a control for the previously reported increase in Cer synthesis induced by ABT-263, we also treated cells with ABT/AZD at high doses and observed that they do, indeed, induce increases in Cer species.

### 3.8. CerS 6 Deletion Impairs Transport of ectoCRT to the Cell Surface

Fumonisin B1 treatment blocked cell surface translocation of ectoCRT (Figure 5C) indicating that Cer was required for ectoCRT transport. However, our sphingolipidomic analysis indicated that ABT/AZD + PF-543 did not significantly induce Cer accumulation. This suggests that basal levels of Cer species are sufficient for ectoCRT transport. Furthermore, this also indicates that the process that drives ectoCRT transport to the cell surface is a normal cellular function, not one that is directly caused by Cer accumulation. Nevertheless, we next wanted to determine which, if any, Cer species were involved in the process of ectoCRT transport to the cell surface.

Given that ICD is a form of regulated cell death and that C16:0 Cer has long been associated with apoptosis and other forms of cell death, we focused on the effects of C16:0 Cer and the Cer synthases responsible for its production (CerS5 and CerS6). To this end, we generated CerS6 knock-out cell lines in DLD-1 cells using Crispr/Cas9. As shown in Figure 8A, individual clones were confirmed to have deletion of the CerS6 protein and one clone was selected for further evaluation. To confirm that CerS6 KO altered levels of C16:0 Cer production, we conducted sphingolipidomic analysis in wild-type DLD-1 and CerS6 KO clones. As shown in Figure 8B, we observed a significant reduction in levels of C16:0 and dhC16:0 Cer indicating that CerS6 is the Cer synthase primarily responsible for the production of C16:0 in DLD-1 cells. The normalized average level of the C16:0 Cer species decreased from 0.85 (pmoles/nmolePi) for wild-type DLD-1 cells to 0.28 (pmoles/nmolePi) for the CerS6KO DLD-1 cells (95% CI 0.054 to 1.09, *p* = 0.029) and the C16:0 dhCer species decreased from 0.94 (pmoles/nmolePi) for wild-type DLD-1 cells to 0.17 (pmoles/nmolePi) for CerS6KO cells (95% CI 0.26 to 1.29, *p* = 0.0041).

We also generated and selected a CerS5 KO clone for comparison to the CerS6 KO clone (Figure 8C). We next examined the cell surface exposure of ectoCRT in wild-type DLD-1 and Cers5 and CerS6 knock-out clones. Interestingly, as shown in Figure 8D, the CerS5 KO had higher levels of ectoCRT, particularly in response to ABT/AZD alone. The average MFI increased from 155 for wild-type DLD-1 cells to 310 for the CerS5Ko (95% CI 102.1 to 207.9, *p* < 0.0001) Conversely, the CerS6 KO had significantly less ectoCRT exposure. The average MFI decreased from 155 for wild-type DLD-1 cells to 53.3 for the CerS6KO (95% CI 48.81 to 154.7, *p* = 0.0002). The addition of PF-543 altered the overall levels of ectoCRT exposure of wild-type, CerS5KO and CerS6KO DLD-1 cells, but did not alter the significant differences between the cell lines. The average MFI increased from 509.6 for wild-type DLD-1 cells to 596.1 for the CerS5KO in response to ABT/AZD + PF-543 (difference of 86.5 MFI, 95% CI 33.5 to 139.4, *p* = 0.0012). Again, the average MFI decreased from 509.6 for wild-type DLD-1 cells to 175.5 for the CerS6KO (difference of 334.1 MFI, 95% CI 281.2 to 387.07, *p* < 0.0001). Since both CerS5 and CerS6 can produce C16:0 Cer, we considered the possibility that CerS6 may be up-regulated in CerS5 KO cells. As shown in Figure 8E, CerS6 protein levels are increased in the CerS5 KO cell line relative to wild-type DLD-1 cells. This suggests that C16:0 Cer, produced by CerS6, is required for transport of ectoCRT to the cell surface. This observation is in line with our data that inhibition of Cer synthesis blocked ectoCRT exposure without affecting dimerization of intracellular calreticulin.

To exclude the possibility that CerS6 KO might affect early events in the ICD process, we conducted a time course analysis of wild-type DLD-1 and CerS6 KO cells treated with vehicle (V), low dose ABT/AZD (L), low dose ABT/AZD + PF-543 (C) and high dose ABT/AZD (H). Comparing the wild-type DLD-1 and CerS6KO cell lines indicates that C16:0 Cer reduction does not affect the SphK1 inhibitor-induced depletion of c-FLIP, nor the activation of Casp8 as judged by the cleavage of its substrate Bap31 (Figure 8F). Dimerization of intracellular calreticulin also proceeded with the same kinetics in wild-type and CerS6KO cells. Together, these data indicate that C16:0 Cer is selectively involved in the process of transport of ectoCRT from its intracellular source to the plasma membrane.

### 3.9. SphK Inhibition Enhances ABT/AZD-Induced Phagocytic Uptake of Dying DLD-1 Cells

EctoCRT is an “eat-me” signal that enhances the phagocytic uptake of dying cancer cells by cells of the innate immune system [3]. To determine if ABT/AZD-induced ectoCRT exposure translates to a functional induction of ICD, we developed a phagocytosis assay. The human acute myeloid leukemia cell line, THP-1, was differentiated into M1 macrophage-like cells according to standard protocols. Wild type DLD-1 cells were treated with PF-543 and ABT/AZD alone and in combination, for 24 h. DLD-1 cells were collected, labeled and co-cultured with counter-labeled THP-1 macrophages for 18 h. Co-cultured cells were visualized by time-lapsed fluorescent microscopy. Vehicle and PF-543 treatment did not induce phagocytosis of DLD-1 cells as judged by the lack of red:green colocalization (Figure 9A). Analysis of multiple fields-of-view, at 18 h, indicate that ABT/AZD alone induced engulfment of DLD-1 cells (yellow cells; white arrows). ABT/AZD + PF-543 enhanced phagocytosis relative to ABT/AZD alone, consistent with an enhancement of ectoCRT exposure.

Engulfment of cells undergoing ICD, by DC/APC/macrophages, induces the production of pro-inflammatory cytokines. Hence, conditioned media from wild-type DLD-1/THP-1 macrophage co-cultures were examined for changes to cytokine profiles by Proteome Profiler Human Cytokine Array (R&D Systems). As shown in Figure 9B, ABT/AZD induced and the addition of PF-543 significantly enhanced a reduction in levels of GM-CSF and an increase in levels of G-CSF and TNFα. The average fold decrease in GM-CSF levels between vehicle and ABT/AZD + PF-543 treatments was 0.83-fold (95% CI, 0.41 to 1.25, *p* = 0.0004). Conversely, G-CSF increased an average of 0.57-fold (95% CI, 0.16 to 0.99, *p* = 0.0073) and TNFα increased an average of 0.45-fold (95%CI, 0.027 to 0.86, *p* = 0.036) in ABT/AZD + PF-543 treated cells relative to vehicle. These data indicate the ability of THP-1 macrophages to recognize and phagocytose ICD cells due to ABT/AZD + PF-543 treatment.

### 3.10. ABT/AZD + PF-543 Induces Bona Fide Immunogenic Cell Death in Syngeneic Mouse CRC Models

To demonstrate the ability of ABT/AZD + PF-543 to induce *bona fide* ICD, we performed the gold-standard “vaccination assay” [33] in a syngeneic, immunocompetent mouse model of CRC. Because immunocompromised mice are required to xenograft human cancer cells, the ability of agents to induce ICD in human cells cannot be characterized in vaccination assays [34,35,36]. As detailed in Figure 9C, MC-38 CRC “vaccine cells” were treated singly with vehicle, PF-543, mitoxantrone (MTX; positive ICD control) or the combination of ABT/AZD + PF-543 for 24 h. Vaccination of injured/dying cells was performed by sub-cutaneous injection into one flank of C57/BL6-albino mice. After 1 week, the vaccinated mice were challenged with live MC-38 CRC cells, injected into the contra-lateral flank, to monitor tumor formation. All mice from each treatment group were followed for survival endpoints and animals showing signs of distress were humanely euthanized. Kaplan–Meier survival curves were calculated. The degree to which tumor formation was reduced/survival was increased is reflective of the strength of the ICD-induced immunogenic response. As shown in Figure 9C, both the positive control ICD-inducer, MTX (*p* = 0.0279 versus vehicle and *p* = 0.0215 versus PF-543 alone) and ABT/AZD + PF-543 (*p* = 0.0044 versus vehicle and *p*= 0.0071 versus PF-543 alone) significantly increased overall survival of vaccinated mice relative to either vehicle or PF-543 alone control vaccines. These results indicate that the combination of ABT/AZD and PF-543 induces the cell surface exposure of the hallmark marker of ICD ectoCRT and that ectoCRT exposure, along with the production of other DAMPs, is sufficient to induce bona fide ICD in response to ABT/AZD + PF-543.

## 4. Discussion

We recently demonstrated that MTX, a well-characterized inducer of ICD causes Cer accumulation through recycling of Sph to form Cer. Inhibition of the SphKs blocked conversion of Sph to S1P and further enhanced Cer formation and production of dimeric ectoCRT [2]. Elsewhere, we have demonstrated that synthetic cannabinoids induce ICD through induction of de novo synthesis of Cer and that inhibition of the SphKs enhances these effects [37]. Herein, we extend our recent observations that sphingolipid metabolism plays a key role in the induction of ICD and provide evidence for the mechanistic roles of both S1P and C16:0 Cer in the process of ICD.

Inhibition of SphK by itself is non-cytotoxic, however in response to minimal cellular insult (i.e., non-cytotoxic doses of ABT-263 and AZD-5991) SphK activity and the production of S1P becomes critical for cell survival. Our data demonstrate that S1P has pro-survival/anti-ICD function that blocks the earliest steps in the pathway leading to ICD. Inhibition of SphK leads to the depletion of c-FLIP an endogenous inhibitor of Casp8 and induces Casp8 activation in response to ABT/AZD. Importantly, Cer formation was not required for this event indicating that the stabilization of c-FLIP is attributed to S1P.

How S1P stabilizes c-FLIP is unknown at the present time. It is possible that S1P binds directly to c-FLIP. S1P, as a small hydrophobic molecule with a hydrophilic PO_4_ head group, could bind to c-FLIP in such a way as to mimic a phosphorylated amino acid (e.g., phosphoSer, phosphoThr or phosphoTyr). Phosphorylation of amino acids commonly affects the stability of proteins leading to or preventing their proteolytic processing [38].

Another possibility would be that S1P affects the activity of ubiquitin ligases that regulate c-FLIP stability. To this end, c-FLIP has been recently shown to be linearly ubiquitinated (M1-ubiquitnated) and stabilized by the linear ubiquitin chain assembly complex (LUBAC) [39]. The activity of LUBAC has been best studied in regulation of TNFα signaling in which heterotypic K63/M1 chains predominate [40]. In fact, studies suggest that K63 ubiquitin oligomers are the preferred substrate of LUBAC. Interestingly, S1P is a co-factor required for the E3 ubiquitin ligase activity of TRAF2, which preferentially produces K63 polyubiquitin chains [41]. Thus, K63 ubiquitination of c-FLIP, by TRAF2, may be a prerequisite for M1-ubquitnation. In support of this, TRAF2 has been shown to bind to c-FLIP, however, ubiquitination of c-FLIP by TRAF2 was not measured in this study [42].

TRAF2 and SphK1 have recently been identified as interacting partners of the UPR protein IRE1α and S1P was shown to be critical for NF-κB activation is response to ER stress [26]. Consistent with this, we observed that inhibition of IRE1α kinase domain activity completely abrogates SphK1 phosphorylation at Ser225, an event that increases SphK1 catalytic activity above basal levels [32]. As mentioned above, ER localized Bak and Bax have been shown to bind to and induce activation of IRE1α [8]. Therefore, we propose that at low concentrations, ABT/AZD-induced activation of Bak/Bax leads to IRE1α activation and phosphorylation of SphK1. SphK1 interacts with TRAF2, which in turn binds S1P, becomes active and mediates K63-polyubiquitination of c-FLIP that nucleates M1-ubiquitination by LUBAC to stabilize c-FLIP. Through this mechanism, minimal activation of Bak and Bax, by low dose ABT/AZD, makes cells dependent on SphK1/S1P-induced stabilization of c-FLIP for their survival. Inhibition of SphK1/S1P blocks heterotypic K63/M1 ubiquitination of c-FLIP leading to K48-polyubiquitnation and proteasomal degradation and activation of Casp8 (Diagrammed in Figure 10).

Cer accumulation has been associated with induction of apoptosis for decades. Our recent studies suggested that Cer accumulation also has a role in ICD. In this study, we determined that Cer is important for transport of CRT from inside the cell to the cell surface. Specifically, C16:0 Cer seems to be required for the cell surface exposure of ectoCRT, but is not required for intracellular generation of dimerized CRT. Interestingly, in contrast to our previous studies, we did not observe an overall increase in Cer levels after combined treatment with ABT/AZD and PF-543. We believe that this is due to the fact that MTX, and possibly synthetic cannabinoids, induce ER stress and/or ROS species generation in addition to ER Ca^2+^ depletion. ER stress/ROS may stimulate Cer synthesis either through the recycling pathway if SphK activity is inhibited or by stimulating the de novo synthesis pathway. We believe that ABT/AZD, on the other hand, primarily induces ER Ca^2+^ depletion with minimal induction of ER stress/ROS.

Even in the absence of Cer accumulation, our data indicate that C16:0 Cer plays a role in ectoCRT transport to the cell surface. Blocking Cer synthesis (i.e., reducing Cer below basal levels) using FB1 and specifically deleting CerS6 both reduced cell surface exposure of ectoCRT. Thus, endogenous levels of C16:0 Cer are sufficient to allow the translocation of ectoCRT to the cell surface. This indicates that the process by which ectoCRT translocates might be a normal cellular process.

Taken together, our results begin to clarify the roles of S1P and Cer in the process of ICD. Furthermore, they suggest that inhibition of SphK/S1P formation may enhance the efficacy of ICD-inducing agents in vivo. Importantly, this study also demonstrates for the first time that targeting the anti-apoptotic Bcl-2 family proteins Bak and Bax induces ICD. It will be critical to evaluate how induction of ICD affects overall patient response to agents such as Navitoclax (ABT-263) as they proceed toward FDA approval as anti-cancer agents.

Given that the Bcl-2 family protein inhibitors, such as Navitoclax (ABT-263), are currently being tested alone or in combination with other chemotherapeutic agents in solid and hematological tumors, it will be critical to evaluate how induction of ICD affects overall patient response to these agents. Our preliminary observations indicate that ABT-263 alone and in combinations with AZD-5991 and/or PF-543 induces ectoCRT exposure in ovarian, pancreatic, prostate and acute myeloid leukemia cell lines. Thus, targeting the Bcl-2 family proteins may be an effective strategy for induction of ICD, with clinically relevant agents, in a number of cancer types. Further work will be required to determine how different driver mutations affect the ICD-inducing capacity of Navitoclax and whether targeting the sphingolipid metabolic pathway could enhance the anti-tumor response or sensitize resistant tumors to Navitoclax.

Similarly, as clinical trials begin to evaluate induction of ICD as a therapeutic end-point, it will be critical to further examine the effects of Navitoclax and other ICD-inducing agents on particular cell populations within tumors. Whether all cells within the tumor express the antigens recognized by the innate immune system will be especially important. The most critical population of cells to target will be cancer stem cells. For instance, aldehyde dehydrogenase (ALDH) is a marker of cancer stem cells, including colon cancer and the presence of ALDH activity in patient serum has potential as a diagnostic biomarker [43,44,45]. The ability of ICD-inducing agents to activate an innate immune response against these ALDH expressing cancer stem cells could dramatically impact the development of recurrent, drug-resistant CRC tumors. Interestingly, one study demonstrated that ALDH expressing CRC stem cells could be directly isolated and used as an anti-cancer vaccine in a rat syngeneic model, although the vaccine was not prepared using an ICD-inducing agent [46]. Thus, the therapeutic potential of ICD-induction in CRC exists and this study reveals a novel mechanism and clinically relevant targets to induce ICD.

## Figures and Tables

**Figure 1 cancers-14-05182-f001:**
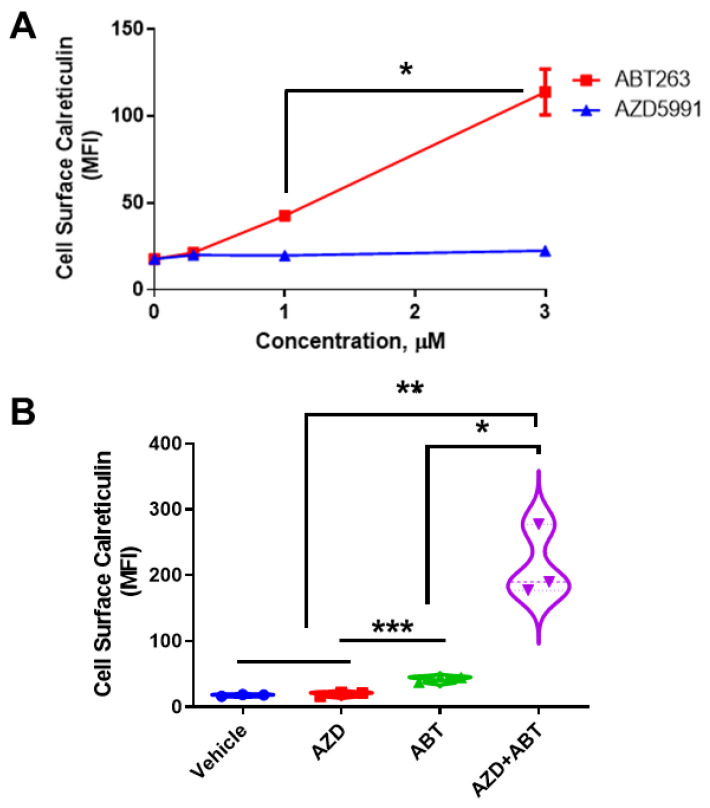
AZD enhances ABT263-induced CRT in DLD-1 cells. (**A**). Surface exposed CRT (ectoCRT) mean fluorescence intensity (MFI) was determined in vehicle treated (DMSO) DLD-1 cells or cells treated with ABT or AZD for 24 h, at the indicated doses, (*n* = 3). Statistical analysis was performed using one-way ANOVA followed by Tukey’s multiple comparison test. (*^,^ **^,^ ***) Asterisks indicate significant changes as reported in results section. (**B**). Surface exposed CRT was determined in untreated (DMSO) DLD-1 cells or cells treated with either ABT (1 µM) or AZD (1 µM) alone or in combination (0.5 µM each) for 24 h, (*n* = 3). Statistical analysis was performed using one-way ANOVA followed by Tukey’s multiple comparison test. (*) Asterisks indicate significant changes as reported in results section. * *p* ≤ 0.05, ** *p* ≤ 0.01, *** *p* ≤ 0.001.

**Figure 2 cancers-14-05182-f002:**
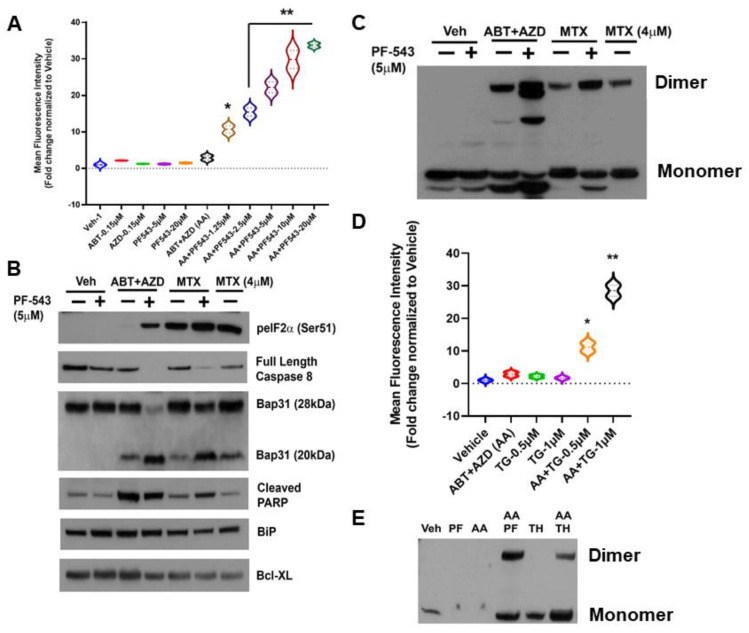
SphK Inhibition enhances the production of a dimeric form of ectoCRT by direct targeting of Bax/Bak with the Bcl-2 family BH3-mimetic inhibitors ABT-263 (Bcl-2 and Bcl-XL) and AZD-5991 (Mcl-1). (**A**). In DLD-1 cells, ABT and AZD alone or in combination had no effect on the cell surface exposure of ectoCRT. PF-543 alone also had no effect on ectoCRT exposure as high as 20 µM. In combination with ABT/AZD, PF-543 dose dependently induces cell surface exposure of ectoCRT. Statistical analysis was performed using one-way ANOVA followed by Tukey’s multiple comparison test. (*^,^ **) Asterisks indicate significant changes as reported in results section. (**B**). To determine whether ABT/AZD ± PF-543 activated Caspase 8, we examined levels of Pro-Caspase 8 (Full length) and cleavage of the Casp8 substrate Bap31. ABT/AZD and MTX alone induce Casp8 activity and Bap31 cleavage. PF-543 enhanced these effects for both ABT/AZD and MTX (*n* = 3). (**C**). ABT/AZD and MTX induce the formation of dimeric CRT (non-reducing conditions) and PF-543 enhances these effects (*n* = 3). (**D**). Activation of Bak/Bax induces pore formation allowing for ER luminal Ca^2+^ leakage to the cytosol. The activity of SERCA can attenuate cytosolic Ca^2+^ levels by replenishing the ER. Inhibition of SERCA with Thapsigargin (TH) enhances ABT/AZD induced ectoCRT exposure. Statistical analysis was performed using one-way ANOVA followed by Tukey’s multiple comparison test. (*^,^ **) Asterisks indicate significant changes as reported in results section. (**E**). Inhibition of SERCA activity enhances ABT/AZD induced dimerization of CRT (*n* = 3). The uncropped blots are shown in Appendix A. * *p* ≤ 0.05, ** *p* ≤ 0.01.

**Figure 3 cancers-14-05182-f003:**
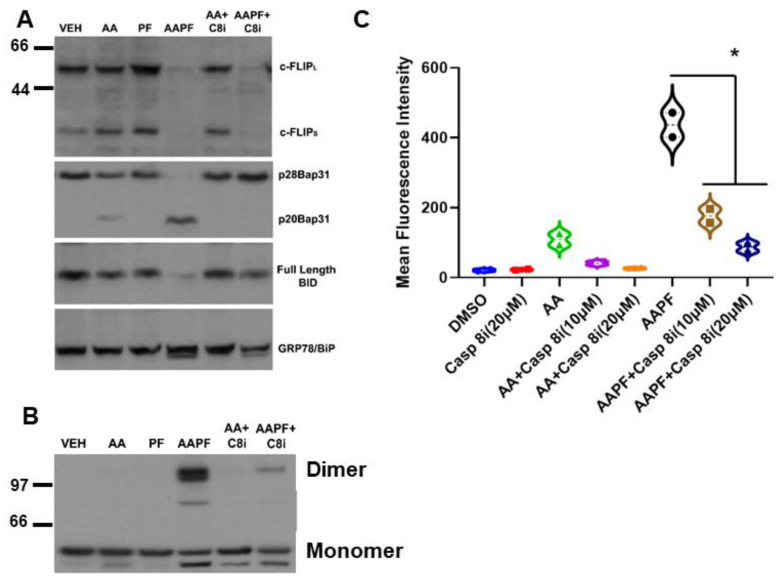
SphK inhibitors synergize with ABT/AZD to induce c-FLIP_L/S_ depletion upstream of Caspase 8 activation. (**A**). DLD-1 cells were treated with ABT/AZD (AA; 0.5/0.25 µM, respectively) ± PF-543 (5 µM) for 16 h. AAPF induced depletion of c-FLIP_L/S_ and induced activation of Caspase 8 as determined by cleavage of the Casp8 substrates Bap31 and BID. Inhibition of Casp8 (z-IETD-fmk; 10 µM) blocked Bap31 and BID cleavage but c-FLIP_L/S_ depletion was independent of Casp8 activity suggesting that SphK generated S1P stabilizes c-FLIP under conditions of ER stress (*n* = 3). (**B**). ABT/AZD + PF-543 induced dimerization of CRT was also blocked by Casp8 inhibition (dimeric and monomeric forms of CRT are indicated). (**C**). Casp8 inhibition dose-dependently reduced ABT/AZD alone and significantly reduced ABT/AZD + PF-543 induced ectoCRT exposure. Statistical analysis was performed using one-way ANOVA followed by Tukey’s multiple comparison test. (* *p* ≤ 0.05) Asterisks indicate significant changes as reported in results section. The uncropped blots are shown in Appendix A.

**Figure 4 cancers-14-05182-f004:**
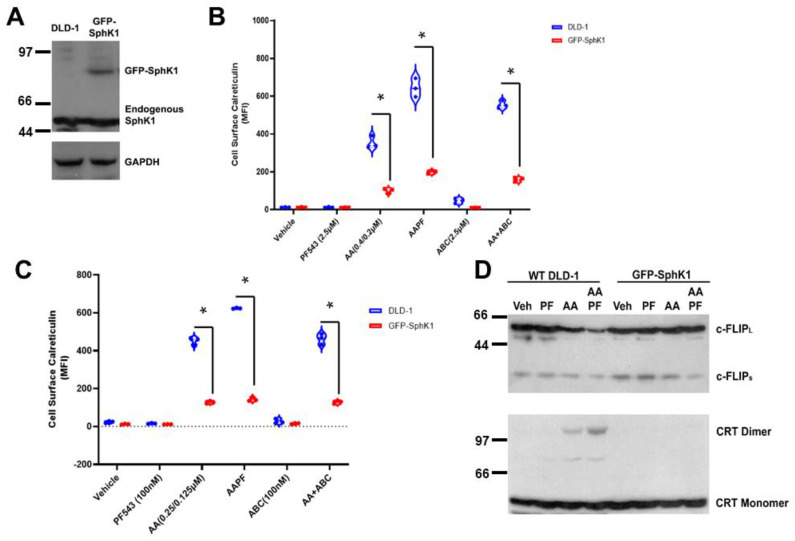
SphK1/S1P over-expression renders DLD-1 cells resistant to ABT/AZD induced ICD by stabilizing c-FLIP. (**A**). Western blot analysis of SphK1 expression using SphK1 Abs. GAPDH control. (**B**). EctoCRT exposure (MFI) under 5% FBS conditions in DLD-1 and GFP-SphK1 cells treated 20 h. Statistical analysis was performed using two-way ANOVA. (* *p* ≤ 0.05) Asterisks indicate significant changes as reported in results section. (**C**). EctoCRT exposure (MFI) under 5% dlFBS conditions in DLD-1 and GFP-SphK1 cells treated 20 h. Statistical analysis was performed using two-way ANOVA. (* *p* ≤ 0.05) Asterisks indicate significant changes as reported in results section. (**D**). Western blot analysis of DLD-1 cells and GFP-SphK1 cells treated with ABT/AZD (0.25/0.125 µM) and PF-543 (100 nM) in 5% dlFBS for 24 h. Depletion of c-FLIP and dimerization of CRT detected by Western blot. The uncropped blots are shown in Appendix A.

**Figure 5 cancers-14-05182-f005:**
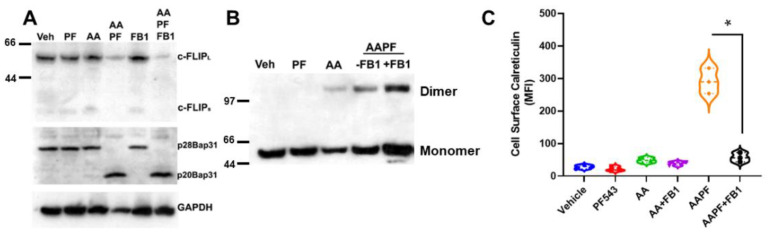
Cer is required for cell surface exposure of ectoCRT. (**A**). Inhibition of Cer synthesis using fumonisin B1 (FB1; 70 µM) had no effect on ABT/AZD + PF-543 induced depletion of c-FLIP and cleavage of the Casp8 substrate, Bap31. (**B**). FB1 treatment enhanced dimerization of cell associated CRT in response to ABT/AZD + PF-543 treatment. (**C**). FB1 significantly reduced cell surface exposure of ectoCRT induced by ABT/AZD + PF-543 treatment. Statistical analysis was performed using one-way ANOVA followed by Tukey’s multiple comparison test. (* *p* ≤ 0.05) Asterisks indicate significant changes as reported in results section. The uncropped blots are shown in Appendix A.

**Figure 6 cancers-14-05182-f006:**
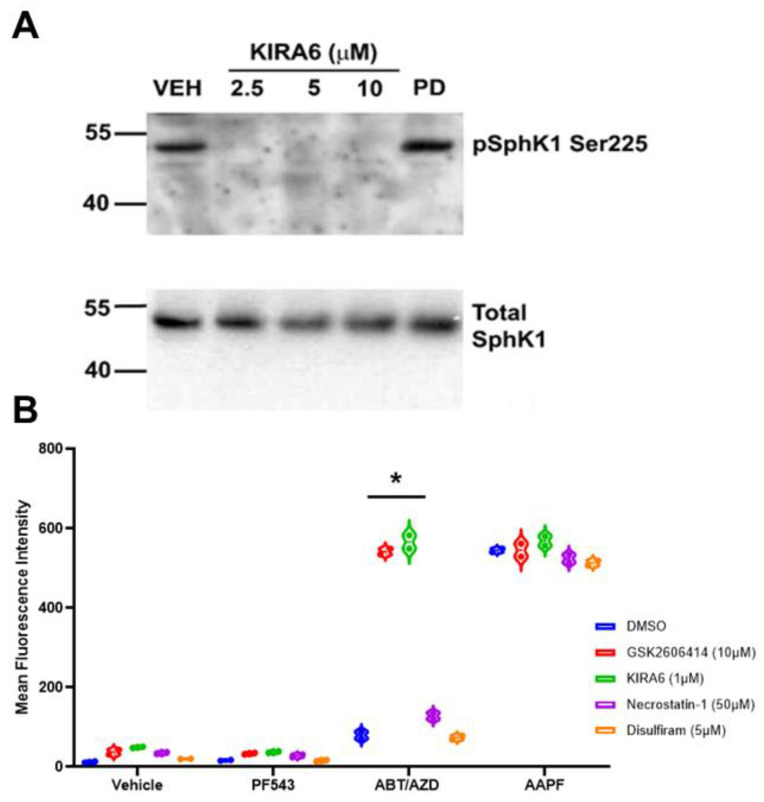
Inhibition of IRE1α blocks activating phosphorylation of SphK1 at Ser225 in a HEK293 over-expression system. (**A**). HEK293 cell over-expressing FLAG-SphK1 [15] were treated with the IRE1α kinase domain inhibitor KIRA6 at the indicated concentrations. Phosphorylation of SphK1 at Ser225 was determined using phospho-specific S225 antibodies (ECM Bio). IRE1α inhibition completely blocked SphK1 phosphorylation as low as 2.5 µM. Erk1/2 has been previously demonstrated to mediate SphK1 phosphorylation at Ser225 [32], however, PD-98059 (20 µM) had no effect on phosphorylation of SphK1. (**B**). Inhibition of IRE1α (KIRA) and PERK (GSK2606414), two arms of the UPR, enhanced cell surface exposure of ectoCRT. Statistical analysis was performed using one-way ANOVA followed by Tukey’s multiple comparison test. (* *p* ≤ 0.05) Asterisks indicate significant changes as reported in results section. Pyroptosis (Disulfiram) and necroptosis (Necrostatin-1) inhibitors had no effect on exposure of ectoCRT indicating ICD is distinct from these forms of RCD. The uncropped blots are shown in Appendix A.

**Figure 7 cancers-14-05182-f007:**
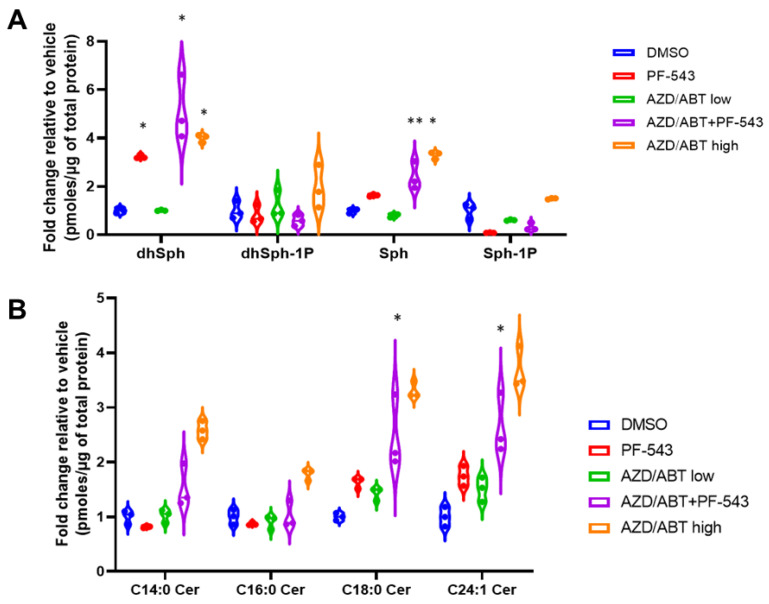
Sphingolipidomic analysis of DLD-1 cells treated with ABT/AZD alone or in combination with PF-543. Sphingolipidomic analysis was conducted on DLD-1 cell samples treated with ABT/AZD low (0.5 µM/0.25 µM) alone and in combination with PF-543 (5 µM) or ABT/AZD high (1.0 µM/0.5 µM) for 24 h, (*n* = 3). Values are presented as pmoles of sphingolipid/µg of total cellular protein. (**A**). Sphingoid-bases and 1-phosphates. (**B**). Select ceramide species. (* *p* ≤ 0.05, ** *p* ≤ 0.01) Asterisks indicate significant changes as reported in results section.

**Figure 8 cancers-14-05182-f008:**
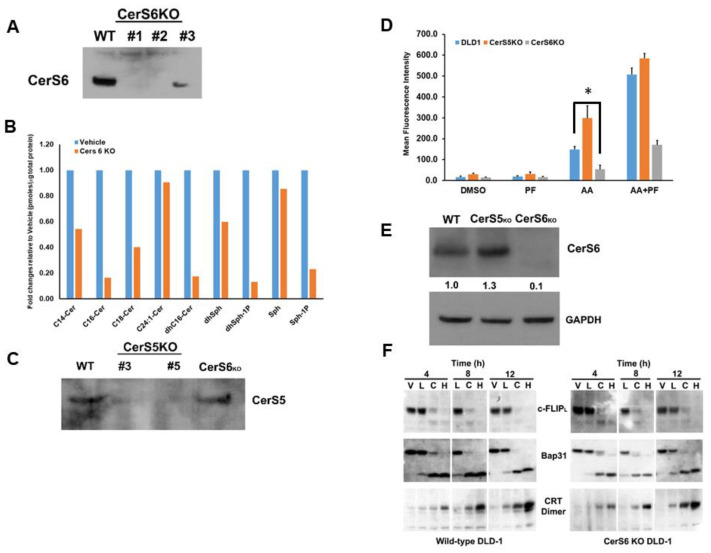
Knock-out of CerS6 attenuates cell surface exposure of ectoCRT. (**A**). Knock-out clones of DLD-1 cells were generated with Crispr/Cas9 gRNAs targeted to CerS6. Knock-out was confirmed by Western blot analysis using anti-CerS6 antibodies. Clone #2 was chosen for further analysis. (**B**). Sphingolipidomic analysis of CerS6 knock-out clone #2. Sphingolipid levels are expressed as average fold change (pmoles/nmolePi) relative to wild-type DLD-1 cells. Statistical analysis was performed using two-way ANOVA. (**C**). Knock-out clones of DLD-1 cells were generated with Crispr/Cas9 gRNAs targeted to CerS5. Knock-out was confirmed by Western blot analysis using anti-CerS5 antibodies. Clone #3 was chosen for further analysis. (**D**). CerS6 knock-out reduced and CerS5 knock-out enhanced cell surface exposure of ectoCRT determined by flow-cytometry (*n* = 3). Statistical analysis was performed using two-way ANOVA. (* *p* ≤ 0.05) Asterisks indicate significant changes as reported in results section. (**E**). Western blot analysis of CerS6 expression in CerS5 and CerS6 knock-out clones normalized to GAPDH. (**F**). Time course analysis. Wild-type DLD-1 and CerS6 knock-out cells were treated for the indicated times with vehicle (V), ABT/AZD low (0.5 µM/0.25 µM; L), ABT/AZD low + PF-543 (5.0 µM; C) and ABT/AZD high (1.0 µM/0.5 µM). Depletion of c-FLIP, cleavage of Bap31 and dimerization of CRT were determined by Western blot analysis using appropriate antibodies. The uncropped blots are shown in Appendix A.

**Figure 9 cancers-14-05182-f009:**
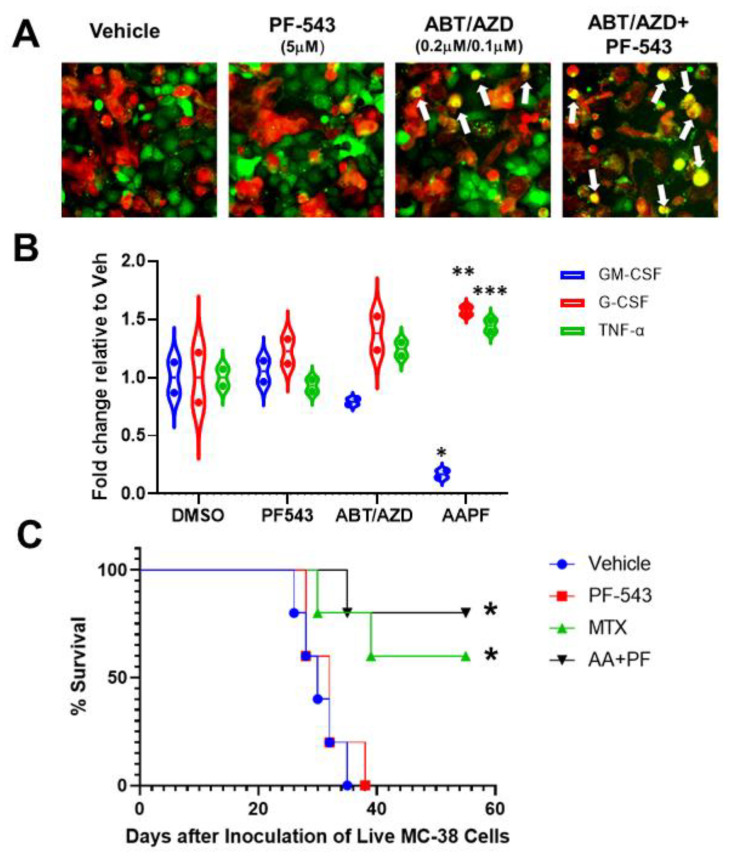
In vitro and in vivo demonstration of induction of ICD by ABT/AZD + PF-543. (**A**). In vitro phagocytic activity assays using THP-1 differentiated M1 macrophages. Representative images of M1 macrophages (red) co-cultured with DLD-1 cells (green) treated as indicated for 24 h White arrows show co-localization of red and green indicative of M1 macrophage engulfment of DLD-1 cells treated with ABT/AZD and ABT/AZD + PF-543. (**B**). Conditioned media from M1 macrophages/DLD1 cells co-culture as described in A were collected and analyzed for expression levels of cytokines associated with ICD. Statistical analysis was performed using one-way ANOVA followed by Tukey’s multiple comparison test. (* *p* ≤ 0.05, ** *p* ≤ 0.01, *** *p* ≤ 0.001) Asterisks indicate significant changes as reported in results section. (**C**). In vivo vaccination assay demonstration of ICD. Mice (*n* = 6/group) were initially vaccinated with 1.0 × 10^6^ injured/dead MC-38 cells treated with mitoxantrone (MTX; 0.3 µM) alone or the combination of ABT/AZD + PF-543 (AA + PF; 0.06 µM/0.13 µM/2.5 µM) for 24 h. Vehicle control cells were subjected to repeated freeze/thaw cycles. 1 week later, vaccinated mice were challenged with live 4.0 × 10^5^ MC-38 cells that were transduced with luciferase, by injection into the contralateral flank. Kaplan–Meier survival curves for mice were generated. Statistical analysis was performed using Log-rank (Mantel-Cox) tests.

**Figure 10 cancers-14-05182-f010:**
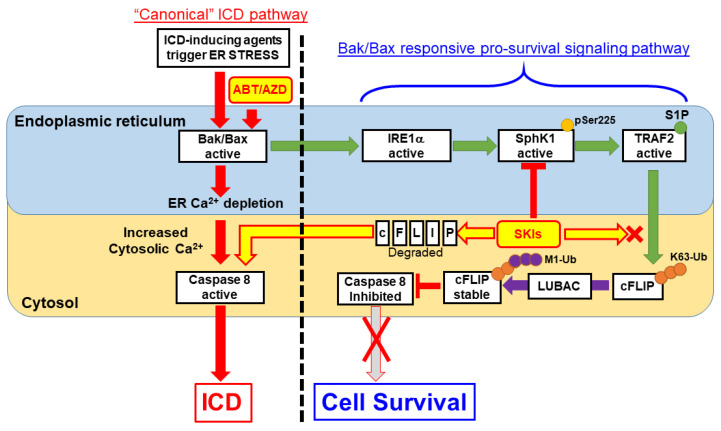
Proposed Model for S1P-mediated pro-survival/anti-ICD signaling. According to our results, ER membrane localized Bak/Bax activation occurs upstream of Caspase 8 activation within the “canonical” ICD pathway. ABT/AZD at high concentrations can directly induce ER lumenal Ca^2+^ depletion and activate Caspase 8 leading to ectoCRT exposure and ICD. At lower, non-cytotoxic concentrations of ABT/AZD, ER membrane localized Bak/Bax induced Caspase 8 activation is counter-balanced by Bak/Bax induced activation of IRE1α kinase activity. IRE1α phosphorylates SphK1 at Ser225 to enhance its catalytic activity above basal levels. SphK1, localized to the cytosolic face of the ER membrane, interacts with TRAF2 and produces S1P locally to activate TRAF2 K63 ubiquitin ligase activity to polyubiquitinate c-FLIP. K63-Ub c-FLIP serves as the substrate for M1 linear ubiquitination, by LUBAC, which stabilizes c-FLIP and inhibits Caspase 8 activity. Thus, induction of ectoCRT dimerization, cell surface exposure and ICD are inhibited. Inhibition of SphK1 activity using SKIs such as PF-543 (among others) blocks S1P production and TRAF2 activation leading to c-FLIP degradation/depletion. Caspase 8 auto-activates leading to ectoCRT dimerization/exposure and ICD.

## Data Availability

Data are contained within the article.

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
