# Peer review of "Regulatory Role of Sphingosine-1-Phosphate and C16:0 Ceramide, in Immunogenic Cell Death of Colon Cancer Cells Induced by Bak/Bax-Activation"

_cancers, 2022, doi:10.3390/cancers14215182_

Round 1
Reviewer 1 Report
The authors have presented a study to prove how sphingolipid response to immunogenic cell death in colorectal cancer, involving the previous knowledge on Bak and Bax on mitochondrial membrane. There are minor questions that are not clear from the manuscript.
1. Sentences are incomplete, please check in the methods section of cell culture.
2. How are you sure that you activated/blocked the ER resident Bak and Bax?
3. How did you confirm there was no ER stress?
4. What is the statistical difference between AZD and ABT in terms of cell surface calreticulin (MFI)

Author Response
REVIEWER 1 Response:
1. Sentences are incomplete, please check in the methods section of cell culture: We have corrected this paragraph
2. How are you sure that you activated/blocked the ER resident Bak and Bax? We do not have direct evidence that ER resident Bak and Bax are targeted. However, as detailed in the results section, the ER is the site of Cer synthesis. We reasoned that an ER resident population (previously demonstrated by Korsmeyer) of Bak/Bax is likely targeted by ABT/AZD since PF-543 enhances the ABT/AZD induced increases in Cer levels.
3. How did you confirm there was no ER stress? As stated in the results section, phosphorylation of eIF2α is an indicator of ER stress. As shown in Figure 2B, ABT/AZD alone does not induce eIF2α phosphorylation (ER stress). The addition of PF-543 does increase ER stress/eIF2α phosphorylation, however, this the levels of eIF2α phosphorylation do not reach those of a known ER stress inducer mitoxantrone (MTX). Thus, we concluded that ER stress is minimally induced. We have altered the statement in the discussion to acknowledge that there is some ER stress induction.
Lines 712 and 713 have been changed to: “We believe that ABT/AZD, on the other hand, primarily induces ER Ca2+ depletion with minimal induction of ER stress/ROS.”
4. What is the statistical difference between AZD and ABT in terms of cell surface calreticulin (MFI). The MFI of ABT treatment is statistically higher than the AZD treatment (p=0.0010). This has been added to Figure 1B. This has been added to Line 190.
Reviewer 2 Report
The study reported by Dr Hengst and colleagues is a well-conceived follow-up to their earlier study in which they observed that inhibition of the sphingosine kinases, SphK1/K2, and production of the phosphorylated product, sphingosine-1-phosphate (S1P), resulted in augmentation of mitoxantrone (MTX)-and cannabinoid-induced immunogenic cell death (ICD) of colorectal cancer cell lines, which was associated with ceramide accumulation and expression of the DAMP, calreticulin (CRT), on the cell surface. The primary objectives of the experimental work described in the current study were to identify the involvement in the process of ICD of the following: i) direct activation of Bak/Bax; ii) sphingolipids, specifically S1P; and iii) accumulation of ceramides and the types thereof.
To address their research questions, these investigators used various cell lines including the DLD-1 (human colorectal cancer), THP-1 cells (M1-like macrophage precursor) and HEK293 cells over-expressing labelled SphK1, as well as CerS5 and CerS6 gene knockouts of DLD-1 cells. They also used a range of pharmacological inhibitors including, but not limited to, ABT-263 (inhibitor of Bcl-2/BclXL), AZD-5991 (inhibitor of Mcl-1), PF-543 (inhibitor of SphK1), KIRA6 (IRE1α inhibitor), GSK2606414 (PERK inhibitor) and fumonisin B1 (inhibitor of ceramide synthesis), as well as specific antibodies for detection of cellular targets including calreticulin, caspase 8, Bap 31, phosphorylated eIF2α, c-FLIP and others. The authors also used a range of molecular, biochemical and immunological procedures to probe the intracellular cascades involved in induction of ICD by either MTX or the combination of ABT/AZD/PF-543.
The results section of the manuscript reflects a detailed progression of the experimental work undertaken by the authors, which is nicely summarized in Figure 10 and although speculative in parts, is certainly plausible. Briefly, two mechanisms were revealed that were dependent on the concentrations of ABT/AZD, these being induction of ICD or pro-survival. In the case of the former, high concentrations of ABT/AZD resulted in Bak/Bax-mediated mobilization of Ca2+ from the ER leading, to activation of caspase 8, translocation of CRT and induction of ICD. At lower concentrations, ABT/AZD was proposed to trigger a series of events involving IRE1α, SphK1 and TRAF2, which result in stabilization of c-FLIP, inhibition of caspase 8 and cell survival, events which are overcome by PF-543. Additional data implicate ceramide synthase 6 and basal levels of C16: ceramide in transporting CRT to the cell surface. Validation experiments revealed that: i) exposure of DLD-1 cells to ABT/AZD/PF-543 resulted in uptake of these damaged cells by THP-1-derived M-type macrophages; and ii) vaccination of C57/BL6-albino mice with MTX-treated MC-38 CRC cells, or with the combination of ABT/AZD/PF-543, resulted in significantly increased survival relative that of mice injected sub-cutaneously with either vector or PF-543 alone, following re-challenge with MC-37 cells.
In my opinion, this is a very well-written and interesting manuscript describing a well- conducted study. I have only one main point: Do the authors believe that the mechanism of induction of ICD, which they have unravelled, is a broadly applicable mechanism of induction of ICD, or is it likely to vary according to the type of cancer and inducer of ICD?
Author Response
REVIEWER 2 Response:
We would like to thank the reviewer for their questions/comments.
Do the authors believe that the mechanism of induction of ICD, which they have unravelled, is a broadly applicable mechanism of induction of ICD, or is it likely to vary according to the type of cancer and inducer of ICD?
We have addressed these important points in a new paragraph added to the end of the discussion. This paragraph begins at Line 730.
Reviewer 3 Report
Although several reviews about the diagnostics of colon cancer have been already published, the discussion on the markers of colon carcinoma in this paper seems to be original. However I have the following suggestions/comments and hope the authors can address them in the review.
Minor revision
1. Some authors showed that the ADH/ALDH activities are higher in tumor cells than in normal colon tissue, suggesting that isoenzymes of ADH may play an important role in carcinogenesis. Among all tested classes of ADH isoenzymes, only class I had higher activity in the serum of patients with colon cancer:
· Jelski Wojciech, Zalewski Bogdan, Chrostek Lech, Szmitkowski Maciej: Alcohol dehydrogenase (ADH) isoenzymes and aldehyde dehydrogenase (ALDH) activity in the sera of patients with colorectal cancer. Clin Exp Med. 2007, 7, 154-157.
· Jelski Wojciech, Mroczko Barbara, Szmitkowski Maciej: The diagnostic value of alcohol dehydrogenase (ADH) isoenzymes and aldehyde dehydrogenase (ALDH) measurement in the sera of colorectal cancer patients. Dig Dis Sci. 2010, 55, 2953-2957
Please discuss (5-6 sentences)
Author Response
REVIEWER 3 Response:
We would like to thank the reviewer for their suggestions that we feel raises interesting new questions not typically dealt with in studies of ICD.
Minor revision
Some authors showed that the ADH/ALDH activities are higher in tumor cells than in normal colon tissue, suggesting that isoenzymes of ADH may play an important role in carcinogenesis. Among all tested classes of ADH isoenzymes, only class I had higher activity in the serum of patients with colon cancer:
Jelski Wojciech, Zalewski Bogdan, Chrostek Lech, Szmitkowski Maciej: Alcohol dehydrogenase (ADH) isoenzymes and aldehyde dehydrogenase (ALDH) activity in the sera of patients with colorectal cancer. Clin Exp Med. 2007, 7, 154-157.
Jelski Wojciech, Mroczko Barbara, Szmitkowski Maciej: The diagnostic value of alcohol dehydrogenase (ADH) isoenzymes and aldehyde dehydrogenase (ALDH) measurement in the sera of colorectal cancer patients. Dig Dis Sci. 2010, 55, 2953-2957
Please discuss (5-6 sentences)
We have discussed the role of ALDH in CRC and added the suggested references in a new paragraphs added to the end of the discussion. This paragraph begins at Line 742.
The role of ALDH as a CRC stem cell marker raises the interesting question of whether ICD/ the innate immune response can directly target cancer stem cells. This will be an important avenue of future exploration as it is critical to ensure that these stem cells are destroyed to prevent recurrence of disease. We thank the reviewer’s for bringing this to our attention.